# Gabapentin in pregnancy and the risk of adverse neonatal and maternal outcomes: A population-based cohort study nested in the US Medicaid Analytic eXtract dataset

Elisabetta Patorno[1]*, Sonia Hernandez-Diaz[2], Krista F. Huybrechts[1], Rishi J. Desai[1], Jacqueline M. Cohen[2], Helen Mogun[1], Brian T. Bateman[1,3]

1 Division of Pharmacoepidemiology and Pharmacoeconomics, Department of Medicine, Brigham and Women's Hospital and Harvard Medical School, Boston, Massachusetts, United States of America, 2 Department of Epidemiology, Harvard T.H. Chan School of Public Health, Boston, Massachusetts, United States of America, 3 Department of Anesthesiology, Perioperative and Pain Medicine, Brigham and Women's Hospital and Harvard Medical School, Boston, Massachusetts, United States of America

* epatorno@bwh.harvard.edu

**Data Availability Statement:** Because of the data use agreement in place, the research team cannot

## Abstract

### Background

Despite the widespread use, only sparse information is available on the safety of gabapentin during pregnancy. We sought to evaluate the association between gabapentin exposure during pregnancy and risk of adverse neonatal and maternal outcomes.

### Methods and findings

Using the United States Medicaid Analytic eXtract (MAX) dataset, we conducted a population-based study of 1,753,865 Medicaid-eligible pregnancies between January 2000 and December 2013. We examined the risk of major congenital malformations and cardiac defects associated with gabapentin exposure during the first trimester (T1), and the risk of preeclampsia (PE), preterm birth (PTB), small for gestational age (SGA), and neonatal intensive care unit admission (NICUa) associated with gabapentin exposure early, late, or both early and late in pregnancy. Gabapentin-unexposed pregnancies served as the reference. We estimated relative risks (RRs) and 95% confidence intervals (CIs) using fine stratification on the propensity score (PS) to control for over 70 confounders (e.g., maternal age, race/ethnicity, indications for gabapentin, other pain conditions, hypertension, diabetes, use of opioids, and specific morphine equivalents). We identified 4,642 pregnancies exposed in T1 (mean age = 28 years; 69% white), 3,745 exposed in early pregnancy only (28 years; 67% white), 556 exposed in late pregnancy only (27 years; 60% white), and 1,275 exposed in both early and late pregnancy (29 years; 75% white). The reference group consisted of 1,744,447 unexposed pregnancies (24 years; 40% white). The adjusted RR for major malformations was 1.07 (95% CI 0.94–1.21, p = 0.33) and for cardiac defects 1.12 (0.89–1.40, p = 0.35). Requiring ≥2 gabapentin dispensings moved the RR to 1.40 (1.03–1.90, p = 0.03) for cardiac defects. There was a higher risk of preterm birth among women exposed to

share the database used for the current paper, which was based on Medicaid, a joint federal and state program that helps provide healthcare coverage for people with low incomes and limited resources in the United States. Other researchers may request to gain access to the Medicaid database through the Research Data Assistance Center (ResDAC) (https://www.resdac.org/).

**Funding:** This study was supported by an R01 grant (R01 MH100216) from the National Institute of Mental Health. EP is supported by a career development grant K08AG055670 from the National Institute on Aging. BTB was supported by a career development grant K08HD075831 from the National Institute Of Child Health & Human Development of the National Institutes of Health. The funders had no role in the design and conduct of the study; collection, management, analysis, and interpretation of the data; and preparation, review, or approval of the manuscript.

**Competing interests:** We have read the journal's policy and the authors of this manuscript have the following competing interests: EP is investigator of an investigator-initiated grant to the Brigham and Women's Hospital from Boehringer Ingelheim, not related to the topic of the submitted work. SH-D has consulted for Boehringer-Ingelheim and UCB for unrelated topics and has worked with the AED pregnancy registry, which is funded by multiple companies. KFH, BTB, and SH-D have been investigators on grants to the Brigham and Women's Hospital from Lilly, GSK, and Pfizer and BTB on grants from Baxalta and Pacira, unrelated to the topic of this manuscript. BTB consults for Aetion for unrelated projects and was a consultant on a postpartum hemorrhage quality improvement project sponsored by a grant from Merck for Mothers. RJD reports grants from Merck, outside the submitted work.

**Abbreviations:** ACE, angiotensin-converting enzyme; ARR, apparent relative risk; CI, confidence interval; FDA, United States Food and Drug Administration; GABA, gamma-aminobutyric acid; hdPS, high-dimensional propensity score; LMP, last menstrual period; MAX, United States Medicaid Analytic eXtract; NICUa, neonatal intensive care unit admission; PS, propensity score; RR, relative risk; SGA, small for gestational age; STROBE, Strengthening the Reporting of Observational Studies in Epidemiology; T1, first trimester.

gabapentin either late (RR, 1.28 [1.08–1.52], $p < 0.01$) or both early and late in pregnancy (RR, 1.22 [1.09–1.36], $p < 0.001$), SGA among women exposed to gabapentin early (1.17 [1.02–1.33], $p = 0.02$), late (1.39 [1.01–1.91], $p = 0.05$), or both early and late in pregnancy (RR, 1.32 [1.08–1.60], $p < 0.01$), and NICU admission among women exposed to gabapentin both early and late in pregnancy (RR, 1.35 [1.20–1.52], $p < 0.001$). There was no higher risk of preeclampsia among women exposed to gabapentin after adjustment. Study limitations include the potential for residual confounding and exposure misclassification.

## Conclusions

In this large population-based study, we did not find evidence for an association between gabapentin exposure during early pregnancy and major malformations overall, although there was some evidence of a higher risk of cardiac malformations. Maternal use of gabapentin, particularly late in pregnancy, was associated with a higher risk of PTB, SGA, and NICUa.

## Author summary

### Why was this study done?

- In addition to being currently US Food and Drug Administration (FDA)-approved for the treatment of partial seizures and postherpetic neuralgia, gabapentin is extensively used off-label for many conditions, including neuropathic pain, fibromyalgia, anxiety, and tremor.

- Despite the increasing number of patients receiving gabapentin prescriptions, little information is available on the safety of this medication during pregnancy.

- We therefore evaluated the association between the use of gabapentin exposure during pregnancy and the risk of a range of neonatal and maternal outcomes.

### What did the researchers do and find?

- We conducted a large population-based cohort study and used several strategies to minimize potential confounding and misclassification of the exposure and the outcome.

- We did not find evidence of an association between gabapentin exposure during the first trimester (T1) of pregnancy and major malformations overall, although there was some evidence of a higher risk of cardiac malformations. There was a higher risk of preterm birth, small for gestational age (SGA), and neonatal intensive care unit admission (NICUa) among women exposed to gabapentin, particularly in late pregnancy.

### What do these findings mean?

- Pregnant women and their physicians should weigh the benefits of treatment with gabapentin with the risks of potential adverse pregnancy outcomes associated with its use.

## Introduction

Gabapentin is a gamma-aminobutyric acid (GABA) analog with GABA agonist activity. In addition to being currently US Food and Drug Administration (FDA)-approved for the treatment of partial seizures and postherpetic neuralgia [1,2] and—in its prodrug version—restless legs syndrome [3], gabapentin is extensively used off-label for many pain conditions, including diabetic neuropathy and other neuropathic pain, fibromyalgia, postoperative pain, anxiety disorders, hot flushes, alcohol withdrawal, and tremor.

Despite the large number of patients receiving gabapentin prescriptions, only sparse information is available on the safety of this medication during pregnancy. The available information on the occurrence of major malformations in the offspring of mothers exposed to gabapentin early in pregnancy appears to rule out large increases in risk, although available studies included small numbers of gabapentin-exposed pregnancies and therefore were not well powered to identify potential smaller teratogenic effects [4–11]. Even fewer data are available on the association between gabapentin and other neonatal or maternal outcomes. Preliminary signals of a potential increase in the risk of selected adverse outcomes, including preterm birth [4, 9, 11], small for gestational age (SGA) [9,11], and admission to the neonatal intensive care unit (NICUa) [9], have been documented, although studies were small and largely did not account for confounding.

Because of the increasing use of gabapentin in many settings of care and the limited information on its safety in pregnancy, there is a critical need for evidence to help pregnant women or women of childbearing age and their healthcare providers to balance the risks and benefits of gabapentin treatment with regard to pregnancy-related outcomes.

To provide evidence on the safety of gabapentin use in pregnancy, we conducted a large cohort study of pregnant women within the US Medicaid Analytic eXtract (MAX) [12] and assessed a range of neonatal and maternal outcomes.

## Methods

### Source of data and study population

Using the MAX dataset, we collected data for 46 US states and the District of Columbia during the period January 2000 through December 2013. Montana and Connecticut were excluded because of difficulty in linking data for mothers and infants, Michigan was excluded because of incomplete data, and data from Arizona were not available. The details of the strategy used to build the study cohort have been previously reported [13]. The study population included all pregnancies resulting in live births among Medicaid-enrolled women 12 to 55 years old, who had continuous eligibility in Medicaid starting from three months prior to the estimated last menstrual period (LMP) to one month after delivery. The date of LMP was estimated based on the date of delivery combined with diagnostic codes for preterm delivery, using a validated algorithm [14]. Continuous Medicaid eligibility was also required among the linked infants for a minimum of three months after birth, unless they died, in which case a shorter eligibility period was allowed. Pregnancies with a documented chromosomal abnormality and pregnancies with exposure to acknowledged teratogenic agents during the first trimester (T1) were excluded (Fig 1). A description of our planned analyses is included in the supplemental material (S1 text). This study is reported as per the Strengthening the Reporting of Observational Studies in Epidemiology (STROBE) guideline (S1 STROBE Checklist).

## Exposure definition

We created four different exposure groups to match the potentially etiologically relevant exposure windows for the outcomes of interest. T1 exposure was defined as pregnancies with at least one filled prescription for gabapentin during the first 90 days of pregnancy (starting from the date of LMP), independently of exposure to gabapentin later in pregnancy. Exposure early in pregnancy was defined as at least one gabapentin dispensing in the first 140 days of pregnancy and no dispensing between the 141st and 245th days. Exposure late in pregnancy was defined as at least one gabapentin dispensing between the 141st and 245th days of pregnancy and no dispensings in the first 140 days. Exposure both early and late in pregnancy was defined as at least one gabapentin dispensing in the first 140 days of pregnancy and at least one dispensing between the 141st and 245th days. For all exposure groups, the reference group consisted of pregnancies with no gabapentin dispensings from 3 months before the start of pregnancy, in order not to misclassify as unexposed women who still had medications from an earlier dispensing available at the start of pregnancy, through the date of delivery.

## Outcomes

For T1 exposure to gabapentin, the primary outcome was the presence of a major congenital malformation in the infant, defined on the basis of inpatient or outpatient ICD-9 diagnostic and procedural codes in the maternal (first month after delivery) [15] or infant (first three months after birth) records. Secondary outcomes included cardiac malformations, the most common malformation type [16]. Other malformations were evaluated as exploratory outcomes (see S1 Table for detailed information on specific outcome definitions).

For gabapentin exposure in early, late, and both early and late pregnancy, we evaluated preeclampsia, preterm birth, SGA, and NICU admission (see S2 Table for specific definitions). For the definition of preeclampsia, preterm birth, and SGA, we used previously validated definitions [16,17].

## Covariates

We considered a large number of potential confounders: maternal age at delivery, race/ethnicity, year of delivery, Medicaid eligibility group, multiple gestation, indications for gabapentin (e.g., epilepsy or seizure, neuropathic pain), other pain conditions, psychiatric disorders, other medical conditions, smoking and other lifestyle factors, concomitant use of medications (e.g., opioids and specific morphine equivalents, other anticonvulsant medications, angiotensin-converting enzyme [ACE] inhibitors), and indicators of disease burden, including the Obstetric Comorbidity Index [18] and measures of healthcare utilization [19] (see S3 Table for a complete list). Race or ethnicity was considered to represent a potentially important confounding factor and was categorized based on information submitted to the Centers for Medicare & Medicaid Services by individual states. For the primary analyses, we measured maternal comorbidities and concomitant medication use during the 3 months before pregnancy through the end of T1. Information deriving from T1 was included in the definition of these covariates, because information on existing medical conditions, which are not expected to be causal intermediates between gabapentin exposure and pregnancy complications, has more opportunity to be recorded during T1 due to the routine checks and examinations related to a new pregnancy. Measures of healthcare intensity (e.g., number of medical visits) were measured only during the 3-month period before pregnancy, in order for these not to be affected by early awareness of possible pregnancy complications.

## Primary analysis

For each exposure group, we defined the prevalence of covariates by exposure group and used standardized differences to evaluate covariate balance between the exposed pregnancies and the reference group [20]. Absolute risks of outcomes and unadjusted risk ratios (RRs) with 95% confidence interval (CI) were calculated. Separately for each exposure group, we estimated propensity scores (PSs) in logistic regression models as the predicted probability of receiving gabapentin conditional upon the previously described covariates. We trimmed the population whose PS fell within the nonoverlapping areas of the PS distributions, and created 50 PS strata according to the distribution of the exposed pregnancies [21]. Due to the large overlap in the PS distribution of gabapentin exposed and unexposed women, all exposed women across the four exposure groups, except for one exposed to gabapentin late in pregnancy, were retained in the analysis after trimming (see S1, S2, S3 and S4 Figs). We calculated weights for the reference group of unexposed pregnancies on the basis of the distribution of the exposed in PS strata and estimated adjusted RR and 95% CI in generalized linear models. We tested the use of the robust variance estimator to account for correlations within women with multiple pregnancies and found it did not appreciably change Cis; thus, we did not include correlation structures in the analyses.

For all analyses presented, results were described as similar to or different from the reference group based on the magnitude of the point estimates, taking into account the precision of each estimate as reflected in the width of its 95% CI. We focused on estimating magnitude of effects in preference to dichotomizing the results as statistically significant or not [22]. Specifically, we judged estimates to be similar to or different from the reference group by three criteria: (1) the strength of the adjusted RR (regardless of whether the 95% CI includes the null), (2) the degree to which the upper bound of the 95% CI indicates low compatibility between the data and a strong adverse effect (i.e., the upper bound of the CI excludes a large increase in the risk of the adverse outcome of concern), and (3) the consistency of the effect estimates across the sensitivity analyses that we conducted. Interpretation of the effect estimates by clinicians and patients to help inform treatment decisions during pregnancy will vary, depending on perceived benefits of treatment, the severity of the adverse outcome, and what is known about the safety of therapeutic alternatives.

## Sensitivity analyses

We conducted several sensitivity analyses to assess the robustness of our findings. First, in order to reduce the potential for exposure misclassification, we updated the exposure definition as filling ≥2 gabapentin prescriptions during each of the previously specified periods of interest, with the assumption that filling multiple prescriptions would increase the likelihood that gabapentin was actually taken or taken more consistently. Second, to ensure that maternal malformations were not being erroneously attributed to the offspring, and thus reduce the chances of outcome misclassification, we redefined the outcome of major malformations using infant claims only and extended infant follow-up to 1 year for infants continuously eligible for Medicaid for ≥1 year. Third, to reduce potential residual confounding due to channeling bias, (1) we used high-dimensional propensity score (hdPS), which enriched the original PS with 100 additional empirically identified covariates [23,24], and (2) we conducted analyses restricted to pregnancies among subgroups of women with either indication of epilepsy or seizures or indication of pain. Fourth, for exposure to gabapentin early, late, and both early and late in pregnancy, to ensure that potential confounders beyond the end of T1 were captured and accounted for, we updated the covariate assessment period by measuring maternal comorbidities, concomitant medication, and healthcare intensity from LMP through the first 140

days of pregnancy. Fifth, to assess the presence of a potential dose-response relationship for gabapentin, we examined the risk of outcomes according to tertiles of the first and the highest prescribed daily dose filled during the specific exposure period of interest. Sixth, for the malformation outcome, as we only included live births in our study cohort, we examined the potential impact of differences in the proportion of terminations among women exposed to gabapentin versus those unexposed on the primary PS-adjusted estimate [25]. Seventh, as maternal smoking has been identified as an important predictor of preterm birth [26], SGA [27], and preeclampsia [28] and is not completely captured in claims data, we quantified the potential impact of residual confounding by smoking on preterm birth, SGA, and preeclampsia in a bias analysis. This quantified the impact of varying the imbalance in the prevalence of maternal smoking between gabapentin-exposed and unexposed pregnancies [29].

## Ethics statement

The research was approved by the Institutional Review Board of Brigham and Women's Hospital. The Institutional Review Board granted a waiver of informed consent (IRB 2013P001741).

# Results

## Study cohort and patient characteristics

Overall, 1,753,865 pregnancies met the inclusion criteria. Among these, 4,642 (0.26%) were exposed to gabapentin during T1, 3,745 (0.21%) were exposed to gabapentin early in pregnancy only (during the first 140 days), 556 (0.03%) were exposed late in pregnancy but not early, and 1,275 (0.07%) were exposed in both early and late pregnancy (Fig 1).

Across all exposure groups, women exposed to gabapentin were older, more frequently white, and had more frequently recorded diagnoses of neuropathic and non-neuropathic pain

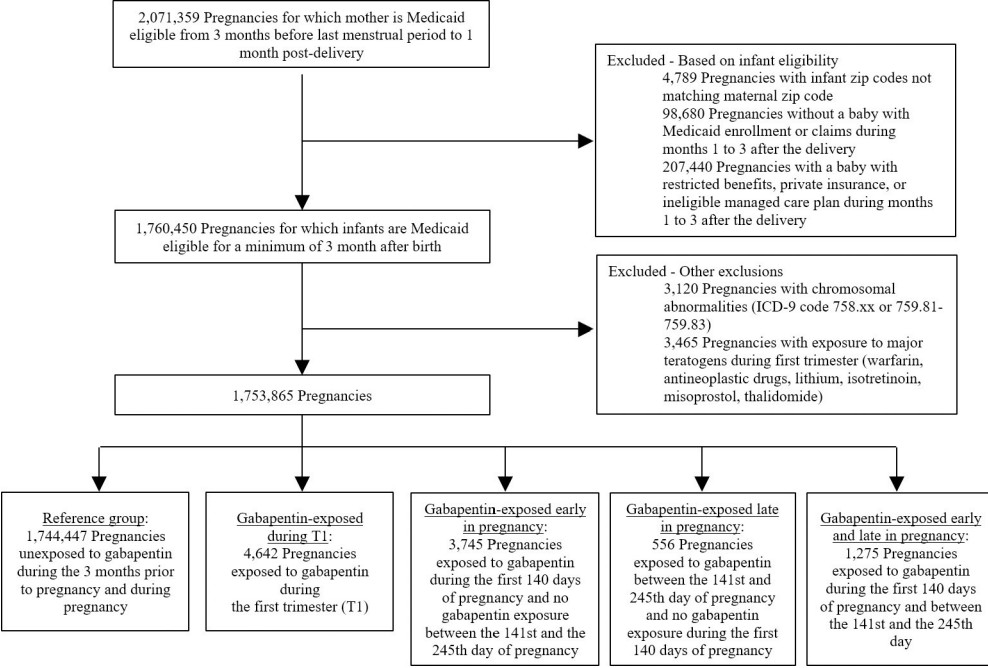

**Fig 1. Flowchart of the study cohort.**

conditions, epilepsy or seizures, and psychiatric disorders, compared with unexposed pregnancies. They had a higher prevalence of hypertension and diabetes and were more likely smokers. They were more frequently users of other anticonvulsant medications, opioids and other pain agents, and psychotropic medications and were characterized by a more pronounced overall burden of disease (Tables 1 and S3). Across all exposure groups, characteristics were well balanced between pregnancies exposed and unexposed to gabapentin after PS adjustment, as evaluated by absolute standardized differences <0.1 [20] (Tables 1 and S4).

## Absolute and relative risks of neonatal and maternal outcomes

The prevalence of overall major congenital and cardiac malformations was 5.0 and 1.9 per 100 live births, respectively, among pregnancies exposed to gabapentin during T1, and 3.3 and 1.1 per 100 among unexposed pregnancies (Table 2). In unadjusted analyses, gabapentin was associated with a higher risk of both overall and cardiac malformations (RR, 1.49 [95% CI 1.31–1.69, $p < 0.001$]; RR, 1.77 [1.43–2.18], $p < 0.001$, respectively) compared to unexposed pregnancies. After PS adjustment, the RR were 1.07 (0.94–1.21, $p = 0.33$), and 1.12 (0.89–1.40, $p = 0.35$), respectively.

Among pregnancies exposed to gabapentin in early, late, and both early and late pregnancy, the prevalence of the other outcomes was 4.9, 5.9, and 6.3 per 100 live births, respectively, for preeclampsia (compared to 4.1 per 100 among unexposed pregnancies), 15.2, 19.2, and 20.2 per 100 live births for preterm birth (compared to 10.5 per 100 among unexposed), 6.1, 6.8, and 7.6 per 100 live births for SGA (compared to 4.9 per 100 among unexposed), and 11.0, 12.2, and 17.6 per 100 live births for NICU admission (compared to 5.8 per 100 among unexposed). Before PS adjustment, gabapentin was associated with an higher risk of preeclampsia (RR, 1.17 [1.01–1.35], $p = 0.03$ for early exposure; RR, 1.43 [1.03–2.00], $p = 0.03$ for late exposure; RR, 1.52 [1.23–1.87], $p < 0.001$ for early and late exposure), preterm birth (RR, 1.46 [1.35–1.57], $p < 0.001$ for early exposure; RR, 1.84 [1.55–2.18], $p < 0.001$ for late exposure; RR, 1.93 [1.73–2.16], $p < 0.001$ for early and late exposure), SGA (RR, 1.71 [1.50–1.96], $p < 0.001$ for early exposure; RR, 1.97 [1.43–2.71], $p < 0.001$ for late exposure; RR, 2.26 [1.85–2.75], $p < 0.001$ for early and late exposure), and NICU admission (RR, 1.89 [1.73–2.07], $p < 0.001$ for early exposure; RR, 2.11 [1.69–2.63], $p < 0.001$ for late exposure; RR, 3.03 [2.69–3.41], $p < 0.001$ for early and late exposure). In PS-adjusted analyses, there was no increase in risk of preeclampsia, regardless of the timing of the gabapentin exposure during pregnancy (RR, 0.87 [0.75–1.00], $p = 0.05$ for early exposure; RR, 0.96 [0.69–1.33], $p = 0.80$ for late exposure; RR, 0.92 [0.74–1.13], $p = 0.42$ for early and late exposure). Conversely, the risks remained elevated for preterm birth among women exposed to gabapentin either late (RR, 1.28 [1.08–1.52], $p < 0.01$) or both early and late in pregnancy (RR, 1.22 [1.09–1.36], $p < 0.001$), for SGA among women exposed to gabapentin early (RR, 1.17 [1.02–1.33], $p = 0.02$), late (RR, 1.39 [1.01–1.91], $p = 0.05$), or both early and late in pregnancy (RR, 1.32 [1.08–1.60], $p < 0.01$), and for NICU admission among women exposed to gabapentin both early and late in pregnancy (RR, 1.35 [1.20–1.52], p<0.001).

## Sensitivity, secondary, and post hoc analyses

Sensitivity and subgroup analyses were largely consistent with the main findings before and after PS adjustment (Tables 3 and S5). After PS adjustment, there was a consistent increase in the risk of preterm birth, SGA, and NICUa associated with gabapentin exposure either late or both early and late in pregnancy, and no increase in risk of overall major malformations or preeclampsia (Table 3). In a few instances, point estimates, although consistent in magnitude and direction of the association, had less precise CIs due to the smaller sample size in analyses

**Table 1. Selected baseline characteristics of gabapentin-exposed and unexposed women before PS adjustment, with standardized differences before and after PS adjustment.**

| Baseline characteristics | Reference: Unexposed (N = 1,744,447) | Exposed during T1 (N = 4,642) | St. Diff. before PS adjustment | St. Diff after PS adjustment | Exposed early in pregnancy (N = 3,745) | St. Diff. before PS adjustment | St. Diff after PS adjustment | Exposed late in pregnancy (N = 556) | St. Diff. before PS adjustment | St. Diff after PS adjustment | Exposed early and late in pregnancy (N = 1,275) | St. Diff. before PS adjustment | St. Diff after PS adjustment |
|---|---|---|---|---|---|---|---|---|---|---|---|---|---|
| Age at delivery, mean (SD) | 24.3 (5.8) | 28.4 (6.0) | 0.69 | 0.00 | 28.1 (6.1) | 0.64 | 0.00 | 27.2 (5.8) | 0.50 | 0.00 | 28.9 (5.8) | 0.79 | 0.01 |
| Race/ethnicity, N (%) | | | | | | | | | | | | | |
| White | 698,670 (40.1) | 3,210 (69.2) | 0.61 | −0.05 | 2,492 (66.5) | 0.55 | −0.04 | 334 (60.1) | 0.41 | −0.02 | 961 (75.4) | 0.77 | −0.05 |
| Black | 569,089 (32.6) | 615 (13.3) | −0.47 | 0.03 | 561 (15.0) | −0.42 | 0.03 | 116 (20.9) | −0.27 | 0.01 | 125 (9.8) | −0.58 | 0.04 |
| Hispanic | 250,779 (14.4) | 317 (6.8) | −0.25 | 0.02 | 294 (7.9) | −0.21 | 0.02 | 55 (9.9) | −0.14 | 0.01 | 53 (4.2) | −0.36 | 0.03 |
| Other[1] | 225,909 (13.0) | 500 (10.8) | −0.07 | 0.01 | 398 (10.6) | −0.07 | 0.01 | 51 (9.2) | −0.12 | 0.01 | 136 (10.7) | −0.07 | 0.02 |
| Multiple gestation, N (%) | 59,636 (3.4) | 219 (4.7) | 0.07 | 0.00 | 177 (4.7) | 0.07 | 0.00 | 34 (6.1) | 0.13 | −0.01 | 59 (4.6) | 0.06 | 0.00 |
| **Labeled indications, N (%)** | | | | | | | | | | | | | |
| Epilepsy or seizures | 11,861 (0.7) | 347 (7.5) | 0.35 | 0.02 | 193 (5.2) | 0.27 | 0.02 | 36 (6.5) | 0.32 | 0.06 | 179 (14.0) | 0.53 | 0.06 |
| Neuropathic pain | 21,701 (1.2) | 1,116 (24.0) | 0.73 | 0.08 | 918 (24.5) | 0.74 | 0.08 | 65 (11.7) | 0.43 | 0.04 | 264 (20.7) | 0.65 | 0.05 |
| Restless legs syndrome | 345 (0.0) | 37 (0.8) | 0.12 | 0.03 | 24 (0.6) | 0.11 | 0.02 | <11 | 0.05 | 0.01 | 15 (1.2) | 0.15 | 0.04 |
| **Pain conditions, N (%)** | | | | | | | | | | | | | |
| Fibromyalgia | 14,545 (0.8) | 397 (8.6) | 0.37 | 0.04 | 332 (8.9) | 0.38 | 0.04 | 35 (6.3) | 0.30 | 0.03 | 91 (7.1) | 0.33 | 0.02 |
| Arthritis, arthropathies, and musculoskeletal pain | 146,564 (8.4) | 1,862 (40.1) | 0.80 | 0.01 | 1,518 (40.5) | 0.81 | 0.01 | 169 (30.4) | 0.58 | 0.00 | 482 (37.8) | 0.74 | 0.00 |
| Back and neck pain | 135,116 (7.8) | 2,099 (45.2) | 0.94 | 0.00 | 1,681 (44.9) | 0.93 | 0.00 | 162 (29.1) | 0.57 | 0.00 | 567 (44.5) | 0.92 | −0.01 |
| Migraine or headache | 124,860 (7.2) | 1,194 (25.7) | 0.52 | −0.01 | 975 (26.0) | 0.52 | −0.01 | 111 (20.0) | 0.38 | 0.00 | 313 (24.6) | 0.49 | 0.00 |
| **Psychiatric conditions, N (%)** | | | | | | | | | | | | | |
| Depression | 106,658 (6.1) | 1,261 (27.2) | 0.59 | −0.02 | 1,013 (27.1) | 0.59 | −0.02 | 98 (17.6) | 0.36 | 0.00 | 336 (26.4) | 0.57 | −0.02 |
| Bipolar disorder | 21,146 (1.2) | 573 (12.3) | 0.45 | 0.02 | 434 (11.6) | 0.43 | 0.02 | 42 (7.6) | 0.31 | 0.03 | 174 (13.7) | 0.49 | 0.03 |
| Anxiety | 64,514 (3.7) | 1,047 (22.6) | 0.58 | 0.00 | 798 (21.3) | 0.55 | 0.00 | 87 (15.7) | 0.41 | 0.02 | 332 (26.0) | 0.66 | 0.00 |
| **Other maternal conditions, N (%)** | | | | | | | | | | | | | |
| Hypertension | 39,100 (2.2) | 419 (9.0) | 0.30 | 0.01 | 318 (8.5) | 0.28 | 0.00 | 40 (7.2) | 0.24 | 0.00 | 132 (10.4) | 0.34 | 0.02 |
| Diabetes | 27,469 (1.6) | 370 (8.0) | 0.30 | 0.00 | 295 (7.9) | 0.30 | 0.00 | 28 (5.0) | 0.19 | 0.00 | 104 (8.2) | 0.31 | 0.01 |
| **Concomitant use of medications, N (%)** | | | | | | | | | | | | | |
| Opioids and opioid-related treatment in T1 | | | | | | | | | | | | | |
| Codeine | 54,340 (3.1) | 418 (9.0) | 0.25 | −0.02 | 351 (9.4) | 0.26 | −0.02 | 44 (7.9) | 0.21 | −0.03 | 100 (7.8) | 0.21 | −0.02 |
| Hydrocodone | 99,899 (5.7) | 1,505 (32.4) | 0.72 | −0.01 | 1,195 (31.9) | 0.71 | −0.01 | 159 (28.6) | 0.64 | −0.03 | 442 (34.7) | 0.77 | −0.02 |
| Oxycodone | 20,474 (1.2) | 642 (13.8) | 0.49 | 0.04 | 503 (13.4) | 0.48 | 0.04 | 59 (10.6) | 0.41 | 0.02 | 202 (15.8) | 0.54 | 0.03 |
| Morphine equivalents, mg, mean (SD) | 50.6 (544.4) | 1,064.4 (2,662.4) | 0.53 | 0.06 | 978.3 (2,532.6) | 0.51 | 0.06 | 732.2 (2,228.5) | 0.42 | 0.03 | 1,395.5 (3,128.5) | 0.60 | 0.05 |
| Anticonvulsants in T1 | | | | | | | | | | | | | |
| Carbamazepine | 1,671 (0.1) | 74 (1.6) | 0.16 | 0.01 | 29 (0.8) | 0.10 | 0.00 | <11 | 0.06 | 0.01 | 51 (4.0) | 0.28 | 0.04 |
| Phenytoin | 1,484 (0.1) | 55 (1.2) | 0.14 | 0.01 | 31 (0.8) | 0.11 | 0.01 | <11 | 0.12 | 0.02 | 26 (2.0) | 0.19 | 0.02 |
| Topiramate | 3,342 (0.2) | 158 (3.4) | 0.24 | 0.03 | 122 (3.3) | 0.24 | 0.03 | <11 | 0.08 | 0.00 | 47 (3.7) | 0.26 | 0.02 |
| Valproate | 3,007 (0.2) | 87 (1.9) | 0.17 | 0.00 | 53 (1.4) | 0.14 | 0.00 | <11 | 0.13 | 0.02 | 39 (3.1) | 0.23 | 0.01 |
| Other anticonvulsants | 8,030 (0.5) | 288 (6.2) | 0.32 | 0.02 | 205 (5.5) | 0.30 | 0.03 | 20 (3.6) | 0.22 | 0.03 | 104 (8.2) | 0.39 | 0.03 |

*(Continued)*

**Table 1.** (Continued)

| Baseline characteristics | Reference: Unexposed (N = 1,744,447) | Exposed during T1 (N = 4,642) | St. Diff. before PS adjustment | St. Diff after PS adjustment | Exposed early in pregnancy (N = 3,745) | St. Diff. before PS adjustment | St. Diff after PS adjustment | Exposed late in pregnancy (N = 556) | St. Diff before PS adjustment | St. Diff after PS adjustment | Exposed early and late in pregnancy (N = 1,275) | St. Diff. before PS adjustment | St. Diff after PS adjustment |
|---|---|---|---|---|---|---|---|---|---|---|---|---|---|
| **Markers of burden of disease** | | | | | | | | | | | | | |
| Obstetric Comorbidity Index[2], mean (SD) | 0.8 (1.4) | 1.7 (2.0) | 0.52 | 0.01 | 1.6 (1.9) | 0.48 | 0.01 | 1.6 (1.8) | 0.45 | −0.01 | 1.9 (2.1) | 0.60 | 0.02 |
| Number of distinct filled prescriptions, mean (SD) | 1.7 (2.4) | 6.1 (4.4) | 1.23 | −0.02 | 5.9 (4.3) | 1.20 | −0.03 | 4.5 (4.1) | 0.82 | −0.02 | 6.3 (4.5) | 1.26 | 0.00 |
| Number of outpatient physician visits, mean (SD) | 2.1 (3.5) | 6.2 (7.4) | 0.70 | 0.04 | 6.1 (7.5) | 0.69 | 0.04 | 4.7 (6.9) | 0.47 | 0.05 | 6.0 (6.7) | 0.72 | 0.01 |
| Patients hospitalized, N (%) | 62,587 (3.6) | 305 (6.6) | 0.14 | 0.02 | 235 (6.3) | 0.12 | 0.02 | 33 (5.9) | 0.11 | 0.02 | 91 (7.1) | 0.16 | 0.02 |
| Number of emergency room visits, mean (SD) | 0.3 (0.9) | 0.9 (1.7) | 0.45 | −0.02 | 0.9 (1.7) | 0.45 | −0.02 | 0.8 (1.5) | 0.38 | −0.02 | 0.9 (1.6) | 0.46 | −0.02 |

Maternal comorbidities and concomitant medication use were measured during the 3 months before pregnancy through the end of T1. Measures of healthcare intensity (e.g., number of medical visits) were measured only during the 3-month period before pregnancy, in order for these not to be affected by early awareness of possible pregnancy complications.

[1]Other race includes Asian, Native American, Other, and Unknown.

[2]The Obstetric Comorbidity Index predicts severe maternal morbidity. The range for the maternal comorbidity index is 0 to 45, with lower values associated with lower burden of maternal illness and higher values associated with higher burden of maternal illness [18].

In accordance with the data use agreement, we do not report information for frequency cells with less than 11 cases. These are noted as <11.

Abbreviations: PS, propensity score; SD, standard deviation; St. Diff., standardized differences, i.e., the difference in means or proportions divided by the pooled standard deviation [20]; T1, first trimester

restricted to subsets of the gabapentin-exposed population, particularly for women with late exposure only. Of note, the risk of cardiac malformations in pregnancies exposed to gabapentin during T1 was increased when we redefined the exposure based on ≥2 filled prescriptions (RR, 1.40 [1.03–1.90], $p$ = 0.03) and in a post hoc analysis with hdPS adjustment (RR, 1.40 [1.03–1.90], $p$ = 0.03), and remained apparently elevated, with a wider CI, in a subgroup analysis that was restricted to patients with a recorded diagnosis of epilepsy or seizures (RR, 1.40 [0.73–2.71], $p$ = 0.31). An additional post hoc analysis, which evaluated specific types of cardiac malformations, revealed an increased risk for conotruncal defects (RR, 1.99 [1.03–3.82], $p$ = 0.04] (S6 Table), which persisted in analyses that were restricted to pregnant women that filled ≥2 prescriptions, redefined the outcome using infant claims only, and with hdPS adjustment. Elevated but imprecisely estimated risks were also observed for left-sided defects and other cardiac defects. Elevated point estimates were observed for central nervous system defects, ear anomalies, and noncardiac vascular defects, albeit with wide CIs (S7 Table). Analyses based on tertiles of the first or the highest gabapentin prescribed daily dose filled during the specific exposure period of interest did not reveal dose-response relations for any of the examined outcomes (S8 Table). After accounting for potential differences in the probability of termination of malformed fetuses among exposed and unexposed women, the range of plausible RRs for major overall malformations estimated for pregnancies exposed to gabapentin during T1 was 1.09 to 1.17 (S5 Fig). Finally, quantification of the bias associated with potential residual imbalances in maternal smoking between exposed and unexposed pregnancies revealed that adjusted RRs of preterm birth, SGA, and preeclampsia for gabapentin exposure

**Table 2. Absolute and relative risk of neonatal and maternal outcomes associated with exposure to gabapentin compared with unexposed pregnancies.**

| Exposure group | Unexposed | Exposed during T1 | Exposed early in pregnancy | Exposed late in pregnancy | Exposed early and late in pregnancy |
|---|---|---|---|---|---|
| **Total** | 1,744,447 | 4,642 | 3,745 | 556 | 1,275 |
| **Outcomes** | | | | | |
| **Major congenital malformations** | | | | | |
| Events | 58,086 | 230 | . | . | . |
| Risk/100 births | 3.3 | 5.0 | . | . | . |
| Unadjusted RR (95% CI), *p*-value | Ref. | 1.49 (1.31–1.69), <0.001 | . | . | . |
| PS-adjusted RR (95% CI), *p*-value | Ref. | 1.07 (0.94–1.21), 0.33 | . | . | . |
| **Cardiac malformations** | | | | | |
| Events | 18,514 | 87 | . | . | . |
| Risk/100 births | 1.1 | 1.9 | . | . | . |
| Unadjusted RR (95% CI), *p*-value | Ref. | 1.77 (1.43–2.18), <0.001 | . | . | . |
| PS-adjusted RR (95% CI), *p*-value | Ref. | 1.12 (0.89–1.40), 0.35 | . | . | . |
| **Preeclampsia** | | | | | |
| Events | 72,197 | . | 182 | 33 | 80 |
| Risk/100 births | 4.1 | . | 4.9 | 5.9 | 6.3 |
| Unadjusted RR (95% CI), *p*-value | Ref. | . | 1.17 (1.01–1.35), 0.03 | 1.43 (1.03–2.00), 0.03 | 1.52 (1.23–1.87), <0.001 |
| PS-adjusted RR (95% CI), *p*-value | Ref. | . | 0.87 (0.75–1.00), 0.05 | 0.96 (0.69–1.33), 0.80 | 0.92 (0.74–1.13), 0.42 |
| **Preterm delivery** | | | | | |
| Events | 182,445 | . | 571 | 107 | 258 |
| Risk/100 births | 10.5 | . | 15.2 | 19.2 | 20.2 |
| Unadjusted RR (95% CI), *p*-value | Ref. | . | 1.46 (1.35–1.57), <0.001 | 1.84 (1.55–2.18), <0.001 | 1.93 (1.73–2.16), <0.001 |
| PS-adjusted RR (95% CI), *p*-value | Ref. | . | 1.00 (0.93–1.08), 0.89 | 1.28 (1.08–1.52), <0.01 | 1.22 (1.09–1.36), <0.001 |
| **SGA** | | | | | |
| Events | 55,803 | . | 205 | 35 | 92 |
| Risk/100 births | 3.2 | . | 5.5 | 6.3 | 7.2 |
| Unadjusted RR (95% CI), *p*-value | Ref. | . | 1.71 (1.50–1.96), <0.001 | 1.97 (1.43–2.71), <0.001 | 2.26 (1.85–2.75), <0.001 |
| PS-adjusted RR (95% CI), *p*-value | Ref. | . | 1.17 (1.02–1.33), 0.02 | 1.39 (1.01–1.91), 0.05 | 1.32 (1.08–1.60), <0.01 |
| **NICU admission** | | | | | |
| Events | 101,202 | . | 411 | 68 | 224 |
| Risk/100 births | 5.8 | . | 11.0 | 12.2 | 17.6 |
| Unadjusted RR (95% CI), *p*-value | Ref. | . | 1.89 (1.73–2.07), <0.001 | 2.11 (1.69–2.63), <0.001 | 3.03 (2.69–3.41), <0.001 |
| PS-adjusted RR (95% CI), *p*-value | Ref. | . | 1.01 (0.93–1.11), 0.77 | 1.21 (0.97–1.51), 0.09 | 1.35 (1.20–1.52), <0.001 |

Abbreviations: CI, confidence interval; NICU, neonatal intensive care unit; PS, propensity score; Ref., reference; RR, risk ratio; SGA, small for gestational age; T1, first trimester

either late in pregnancy or both early and late in pregnancy were fairly robust even under extreme scenarios that assumed maternal smoking prevalence to be up to 2.5-fold higher among gabapentin-exposed compared with unexposed pregnancies (Fig 2).

## Discussion

In a large population-based study, we evaluated neonatal and maternal outcomes in over 4,000 women exposed to gabapentin early in pregnancy and approximately 2,000 women exposed in late pregnancy. We did not find evidence for an association between gabapentin exposure during early pregnancy and major malformations overall, though there was some evidence of a

**Table 3. Sensitivity and secondary analyses for the RRs of neonatal and maternal outcomes associated with exposure to gabapentin compared with unexposed pregnancies after PS adjustment.**

| Exposure group | Unexposed | Exposed during T1 | | Exposed early in pregnancy | | Exposed late in pregnancy | | Exposed early and late in pregnancy | |
|---|---|---|---|---|---|---|---|---|---|
| Outcomes | | PS-adjusted RR (95% CI) | p-Value | PS-adjusted RR (95% CI) | p-Value | PS-adjusted RR (95% CI) | p-Value | PS-adjusted RR (95% CI) | p-Value |
| **Major congenital malformations** | | | | | | | | | |
| Main PS-adjusted analysis | **Ref.** | **1.07 (0.94–1.21)** | **0.33** | . | | . | | . | |
| ≥2 Rx | Ref. | 1.15 (0.95–1.39) | 0.15 | . | | . | | . | |
| Infant claims | Ref. | 1.05 (0.92–1.20) | 0.48 | . | | . | | . | |
| 1-year follow-up[1] | Ref. | 1.05 (0.94–1.18) | 0.37 | | | | | | |
| hdPS-adjusted | Ref. | 1.06 (0.93–1.20) | 0.40 | . | | . | | . | |
| ≥1 epilepsy or seizure dx | Ref. | 1.01 (0.66–1.56) | 0.96 | . | | . | | . | |
| No epilepsy or seizure dx | Ref. | 1.07 (0.93–1.22) | 0.35 | . | | . | | . | |
| ≥1 pain dx[2] | Ref. | 1.06 (0.91–1.22) | 0.48 | . | | . | | . | |
| No pain dx | Ref. | 1.14 (0.87–1.49) | 0.33 | . | | . | | . | |
| **Cardiac malformations** | | | | | | | | | |
| Main PS-adjusted analysis | **Ref.** | **1.12 (0.89–1.40)** | **0.35** | . | | . | | . | |
| ≥2 Rx | Ref. | 1.40 (1.03–1.90) | 0.03 | . | | . | | . | |
| Infant claims | Ref. | 1.13 (0.89–1.44) | 0.33 | . | | . | | . | |
| 1-year follow-up[1] | Ref. | 1.10 (0.92–1.53) | 0.41 | | | | | | |
| hdPS-adjusted | Ref. | 1.11 (0.88–1.40) | 0.37 | . | | . | | . | |
| hdPS-adjusted, ≥2 Rx[3] | Ref. | 1.40 (1.03–1.90) | 0.03 | . | | . | | . | |
| ≥1 epilepsy or seizure dx | Ref. | 1.40 (0.73–2.71) | 0.31 | . | | . | | . | |
| No epilepsy or seizure dx | Ref. | 1.08 (0.84–1.38) | 0.56 | . | | . | | . | |
| ≥1 pain dx[2] | Ref. | 1.07 (0.82–1.40) | 0.61 | . | | . | | . | |
| No pain dx | Ref. | 1.32 (0.84–2.06) | 0.23 | . | | . | | . | |
| **Preeclampsia** | | | | | | | | | |
| Main PS-adjusted analysis | **Ref.** | . | | **0.87 (0.75–1.00)** | **0.05** | **0.96 (0.69–1.33)** | **0.80** | **0.92 (0.74–1.13)** | **0.42** |
| ≥2 Rx | Ref. | . | | 0.98 (0.78–1.23) | 0.88 | 1.17 (0.64–2.14) | 0.60 | 0.87 (0.67–1.14) | 0.32 |
| hdPS-adjusted | Ref. | . | | 0.86 (0.75–1.00) | 0.05 | 0.96 (0.69–1.33) | 0.78 | 1.00 (0.80–1.25) | 0.99 |
| Updated CAP[4] | Ref. | | | 0.88 (0.76–1.02) | 0.08 | 0.98 (0.70–1.36) | 0.89 | 0.93 (0.75–1.15) | 0.51 |
| ≥1 epilepsy or seizure dx | Ref. | . | | 0.74 (0.39–1.41) | 0.37 | 1.43 (0.37–5.46) | 0.60 | 0.82 (0.46–1.46) | 0.49 |
| No epilepsy or seizure dx | Ref. | . | | 0.88 (0.76–1.01) | 0.08 | 0.96 (0.68–1.35) | 0.82 | 0.96 (0.76–1.20) | 0.71 |
| ≥1 pain dx[2] | Ref. | . | | 0.91 (0.77–1.06) | 0.23 | 0.83 (0.52–1.31) | 0.42 | 0.94 (0.74–1.19) | 0.61 |
| No pain dx | Ref. | . | | 0.74 (0.54–1.02) | 0.06 | 1.18 (0.74–1.89) | 0.49 | 0.84 (0.51–1.38) | 0.49 |
| **Preterm delivery** | | | | | | | | | |
| Main PS-adjusted analysis | **Ref.** | . | | **1.00 (0.93–1.08)** | **0.89** | **1.28 (1.08–1.52)** | **<0.01** | **1.22 (1.09–1.36)** | **<0.001** |
| ≥2 Rx | Ref. | . | | 0.98 (0.86–1.12) | 0.81 | 1.27 (0.93–1.74) | 0.14 | 1.28 (1.12–1.46) | <0.001 |
| hdPS-adjusted | Ref. | . | | 1.01 (0.93–1.08) | 0.89 | 1.25 (1.06–1.49) | <0.01 | 1.22 (1.08–1.37) | <0.01 |
| Updated CAP[4] | Ref. | | | 0.99 (0.92–1.06) | 0.74 | 1.24 (1.04–1.47) | 0.01 | 1.20 (1.08–1.34) | <0.01 |
| ≥1 epilepsy or seizure dx | Ref. | . | | 1.07 (0.81–1.42) | 0.62 | 1.14 (0.51–2.51) | 0.75 | 1.47 (1.13–1.91) | <0.01 |
| No epilepsy or seizure dx | Ref. | . | | 1.00 (0.93–1.08) | 0.94 | 1.30 (1.09–1.55) | <0.01 | 1.20 (1.06–1.35) | <0.01 |
| ≥1 pain dx[2] | Ref. | . | | 0.96 (0.88–1.04) | 0.32 | 1.24 (1.00–1.54) | 0.05 | 1.22 (1.08–1.38) | <0.01 |
| No pain dx | Ref. | . | | 1.15 (0.99–1.34) | 0.07 | 1.33 (1.01–1.75) | 0.05 | 1.19 (0.92–1.54) | 0.17 |
| **SGA** | | | | | | | | | |
| Main PS-adjusted analysis | **Ref.** | . | | **1.17 (1.02–1.33)** | **0.02** | **1.39 (1.01–1.91)** | **0.05** | **1.32 (1.08–1.60)** | **<0.01** |
| ≥2 Rx | Ref. | . | | 1.15 (0.92–1.45) | 0.22 | 1.35 (0.74–2.46) | 0.33 | 1.28 (1.00–1.63) | 0.05 |

(*Continued*)

**Table 3.** (Continued)

| Exposure group | Unexposed | Exposed during T1 | | Exposed early in pregnancy | | Exposed late in pregnancy | | Exposed early and late in pregnancy | |
|---|---|---|---|---|---|---|---|---|---|
| Outcomes | | PS-adjusted RR (95% CI) | p-Value | PS-adjusted RR (95% CI) | p-Value | PS-adjusted RR (95% CI) | p-Value | PS-adjusted RR (95% CI) | p-Value |
| hdPS-adjusted | Ref. | . | | 1.17 (1.02–1.33) | 0.02 | 1.36 (0.99–1.88) | 0.06 | 1.39 (1.12–1.72) | <0.01 |
| Updated CAP[4] | Ref. | . | | 1.15 (1.01–1.32) | 0.04 | 1.36 (0.99–1.88) | 0.06 | 1.31 (1.08–1.60) | <0.01 |
| ≥1 epilepsy or seizure dx | Ref. | . | | 1.35 (0.83–2.20) | 0.23 | 1.03 (0.15–7.10) | 0.98 | 1.46 (0.89–2.38) | 0.13 |
| No epilepsy or seizure dx | Ref. | . | | 1.17 (1.01–1.34) | 0.03 | 1.46 (1.05–2.02) | 0.02 | 1.30 (1.05–1.62) | 0.02 |
| ≥1 pain dx[2] | Ref. | . | | 1.16 (1.00–1.35) | 0.05 | 1.33 (0.88–2.01) | 0.18 | 1.29 (1.04–1.61) | 0.02 |
| No pain dx | Ref. | . | | 1.14 (0.85–1.51) | 0.39 | 1.44 (0.86–2.39) | 0.16 | 1.32 (0.84–2.07) | 0.22 |
| **NICU admission** | | | | | | | | | |
| Main PS-adjusted analysis | **Ref.** | **.** | | **1.01 (0.93–1.11)** | **0.77** | **1.21 (0.97–1.51)** | **0.09** | **1.35 (1.20–1.52)** | **<0.001** |
| ≥2 Rx | Ref. | . | | 1.08 (0.93–1.25) | 0.31 | 1.00 (0.64–1.57) | 0.99 | 1.43 (1.24–1.65) | <0.001 |
| hdPS-adjusted | Ref. | . | | 1.01 (0.92–1.11) | 0.80 | 1.19 (0.95–1.49) | 0.12 | 1.31 (1.15–1.50) | <0.001 |
| Updated CAP[4] | Ref. | . | | 1.00 (0.91–1.09) | 0.95 | 1.16 (0.93–1.45) | 0.18 | 1.33 (1.18–1.50) | <0.001 |
| ≥1 epilepsy or seizure dx | Ref. | . | | 0.68 (0.42–1.11) | 0.16 | 0.31 (0.05–2.15) | 0.24 | 1.18 (0.83–1.68) | 0.35 |
| No epilepsy or seizure dx | Ref. | . | | 1.04 (0.95–1.14) | 0.43 | 1.29 (1.03–1.61) | 0.03 | 1.38 (1.22–1.56) | <0.001 |
| ≥1 pain dx[2] | Ref. | . | | 0.95 (0.86–1.06) | 0.38 | 1.13 (0.85–1.50) | 0.41 | 1.30 (1.13–1.48) | <0.001 |
| No pain dx | Ref. | . | | 1.18 (0.99–1.42) | 0.07 | 1.33 (0.94–1.90) | 0.11 | 1.56 (1.21–2.01) | <0.001 |

[1]Restricted to infants continuously eligible for ≥1 year.

[2]Includes neuropathic pain, fibromyalgia, arthritis, arthropathies and musculoskeletal pain, back and neck pain, migraine or headache, osteoarthritis, rheumatoid arthritis, or other pain.

[3]Post hoc analysis.

[4]CAP measured from the LMP through the first 140 days of pregnancy.

Abbreviations: CAP, covariate assessment period; CI, confidence interval; dx, diagnosis; hdPS, high-dimensional propensity score; LMP, last menstrual period; NICU, neonatal intensive care unit; PS, propensity score; Ref., reference; RR, risk ratio; Rx, filled prescription; SGA, small for gestational age; T1, first trimester

higher risk of cardiac malformations. Sensitivity analyses restricted to women that may use gabapentin more consistently suggested there may be an increased risk of cardiac malformations, and a subsequent post hoc analysis, which evaluated individual cardiac malformations, revealed a potential increase in the risk of conotruncal defects specifically. Despite the large attenuations from crude to adjusted results, maternal use of gabapentin late in pregnancy, regardless of its use early in pregnancy, remained associated with an approximately 20% to 30% increased risk of preterm birth and a 30% to 40% increased risk of SGA. We also observed a 35% increased risk in NICU admission among the offspring of women exposed to gabapentin throughout pregnancy.

Our results confirm the findings from previous studies, which excluded large increases in the risk of major malformations associated with maternal use of gabapentin [4,5,7–10]. However, these studies assessed the risk of major malformations in small populations, which included between 31 and 223 gabapentin-exposed pregnancies, mostly among women with epilepsy, and reported rather imprecise RR estimates (ranging from 0.3 to 1.8 for the use of gabapentin) [5,7–10]. Our study population, which included 4,642 pregnancies exposed to gabapentin during T1 and was not limited to women with epilepsy, allowed us to rule out meaningful increases in the risk of overall major malformations among pregnancies exposed to gabapentin during T1 with higher precision, while permitting to identify a potential moderate increase in the risk of cardiac malformations among women that may use gabapentin more

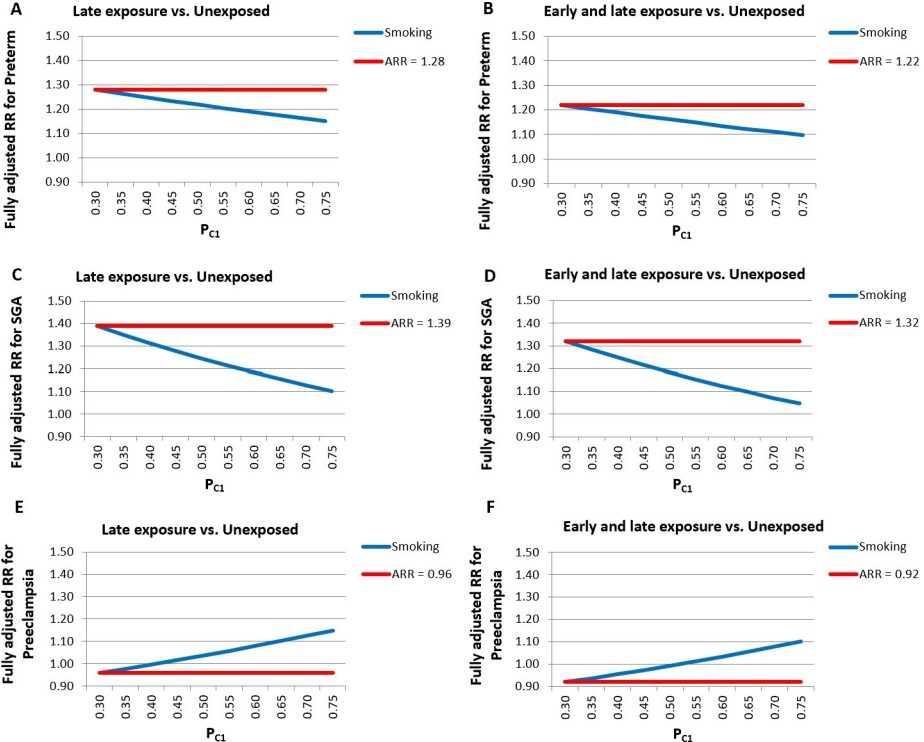

**Fig 2.** Bias analysis quantifying the impact on point estimates of increasing residual differences in the prevalence of maternal smoking between unexposed and gabapentin-exposed women either late (A, C, E) or both early and late in pregnancy (B, D, F). ARR, apparent relative risk; $P_{C1}$, prevalence of maternal smoking among exposed women; RR, relative risk; SGA, small for gestational age

consistently. The specific association observed between gabapentin and conotruncal defects has not been previously reported and should be confirmed or refuted in future investigations.

Our results are also consistent with the limited prior research on the association of gabapentin with other pregnancy-related outcomes. The European Gabapentin Registry study found that among 39 women exposed to gabapentin during pregnancy, the frequency of maternal complications, including eclampsia and SGA, were similar to those observed in the general population, although the frequency of preterm birth was elevated (22.7% versus 11.8%) [4]. A cohort study comparing 223 gabapentin-exposed pregnancies with 223 pregnancies exposed to a nonteratogenic substance described a higher frequency of preterm births (10.5% versus 3.9%), low birth weight (10.5% versus 4.4%), and NICU admission (38% versus 2.9%), but not SGA [9]. Finally, a small population-based study in Italy comparing 11 newborns exposed to gabapentin during pregnancy to unexposed pregnancies found that gabapentin exposure was associated with increased risk of preterm birth (OR = 7.37, 95% CI 1.87–30.54) and SGA (OR = 5.14, 95% CI 1.10–20.23) [11]. The careful adjustment for over 70 potential confounders that was implemented in our study, including detailed accounting for maternal use, timing, and dose of opioids, may explain the reduced magnitude of the associations compared to previous studies, and the large attenuation from our crude to PS-adjusted results, in particular for NICU admission.

This study has limitations. First, certain important patient characteristics (e.g., lifestyle factors and severity of comorbidities) may not be fully captured in claims databases, and this could cause residual confounding that may explain the increased risk for preterm birth, SGA, and NICU admission observed for gabapentin, particularly when used late in pregnancy. However, (1)

additional analyses aimed at mitigating possible residual confounding, i.e., hdPS-adjustment, the restriction to women with either epilepsy/seizures or pain indications, and the extension of the covariate assessment period through the first 140 days of pregnancy, produced results consistent with the main analyses; and (2) an analysis that quantified the potential bias associated with a strong risk factor for preterm birth, SGA, and preeclampsia (protective), revealed that adjusted results were fairly robust even under extreme scenarios of imbalance of maternal smoking prevalence between exposed and unexposed pregnancies. Second, although claims databases include detailed data on filled medications, they do not include information on their actual use by patients, which could lead to drug exposure misclassification. To limit this possibility, in sensitivity analyses we updated the definition of exposure as filling ≥2 gabapentin prescriptions during each of the previously specified periods of interest; these analyses largely confirmed the main findings. Third, the identification of outcomes in claims databases may be affected by outcome misclassification. To reduce this possibility, we used either validated or highly specific definitions of the outcomes [16,17]. We also redefined the malformation outcomes using infant claims only and extended follow-up to 1 year for infants continuously eligible for ≥1 year, which confirmed the primary results. Fourth, because we restricted our study population to live births, spontaneous abortions or therapeutic terminations due to congenital malformations diagnosed early in pregnancy remain unobserved. It has been previously documented that planned terminations may be approximately 10% higher among gabapentin users [9]. A bias analysis, which quantified the potential impact of such missed terminations, suggested that a corrected RR estimate for overall malformations would range between 1.09 and 1.17 depending on the selection probability among the unexposed. Fifth, in the context of multiple analyses, the possibility of a chance finding should be taken into consideration. Finally, our study population included only women who had continuous eligibility in Medicaid starting from three months prior to the estimated LMP to one month after delivery. This may have resulted in the selection of a more disadvantaged subpopulation within Medicaid, mostly composed of low-income adults, multiparae, and women with disabilities, as previously shown [30]. The characteristics of this population of pregnant women, i.e., young, racially diverse, with a high burden of disabilities, are not expected to affect the biological relations evaluated in this study. Therefore, our findings should generalize to other populations.

Our results add to the current understanding of the safety of gabapentin prenatal use and provide pregnant women with pain conditions and epilepsy and their providers with important information, which can guide clinical decisions during pregnancy. Our findings also suggest that pregnant women using gabapentin during pregnancy may be considered for targeted interventions to monitor for and promptly respond to the potential adverse outcomes associated with the use of this agent.

## Conclusions

Results from this large cohort study suggest that gabapentin exposure during early pregnancy does not appear to be associated with teratogenic effects, although a moderately higher risk of cardiac malformations—in particular, conotruncal defects—cannot be excluded. Maternal use of gabapentin, particularly late in pregnancy, was associated with a higher risk of preterm birth, SGA, and NICU admission; an association that was only partially explained by confounders. Clinicians should weigh these potential risks with the clinical benefits of using gabapentin to treat painful and disabling conditions.

## Supporting information

**S1 Table. Definitions for congenital malformations.**
(DOCX)

**S2 Table. Definitions for preeclampsia, preterm birth, SGA, and NICUa.** NICUa, neonatal intensive care unit admission; SGA, small for gestational age.
(DOCX)

**S3 Table. Baseline characteristics of gabapentin-exposed and unexposed women, before PS adjustment.** PS, propensity score.
(DOCX)

**S4 Table. Baseline characteristics of gabapentin-exposed and unexposed women, after PS adjustment.** PS, propensity score.
(DOCX)

**S5 Table. Sensitivity and secondary analyses for the RR of neonatal and maternal outcomes associated with exposure to gabapentin compared with unexposed pregnancies before and after PS adjustment.** PS, propensity score; RR, relative risk.
(DOCX)

**S6 Table. Post hoc analyses for the RR of specific types of cardiac malformations associated with exposure to gabapentin during T1 compared with unexposed pregnancies.** RR, relative risk; T1, first trimester.
(DOCX)

**S7 Table. Sensitivity and secondary analyses for the RR of individual noncardiac major malformation groups associated with exposure to gabapentin compared with unexposed pregnancies.** RR, relative risk.
(DOCX)

**S8 Table. RR of cardiac malformations comparing gabapentin-exposed to unexposed women, stratified by dose tertiles of the first and the highest prescription filled during each exposure period of interest.** RR, relative risk.
(DOCX)

**S1 Fig. PS distributions of gabapentin-exposed and unexposed women during the T1, after PS trimming and weighting.** PS, propensity score; T1, first trimester.
(TIF)

**S2 Fig. PS distributions of gabapentin-exposed and unexposed women early in pregnancy, after PS trimming and weighting.** PS, propensity score.
(TIF)

**S3 Fig. PS distributions of gabapentin-exposed and unexposed women late in pregnancy, after PS trimming and weighting.** PS, propensity score.
(TIF)

**S4 Fig. PS distributions of gabapentin-exposed and unexposed women early and late in pregnancy, after PS trimming and weighting.** PS, propensity score.
(TIF)

**S5 Fig. Corrected RR for the association between gabapentin exposure during the T1 and major malformations.** RR, relative risk; T1, first trimester.
(TIF)

**S1 Text. Planned analyses.**
(DOCX)

**S1 STROBE Checklist. STROBE, Strengthening the Reporting of Observational Studies in Epidemiology.**
(DOCX)

## Author Contributions

**Conceptualization:** Elisabetta Patorno, Sonia Hernandez-Diaz, Krista F. Huybrechts, Brian T. Bateman.

**Formal analysis:** Elisabetta Patorno, Sonia Hernandez-Diaz, Krista F. Huybrechts, Rishi J. Desai, Jacqueline M. Cohen, Helen Mogun, Brian T. Bateman.

**Funding acquisition:** Sonia Hernandez-Diaz, Krista F. Huybrechts.

**Investigation:** Elisabetta Patorno, Sonia Hernandez-Diaz, Krista F. Huybrechts, Rishi J. Desai, Jacqueline M. Cohen, Helen Mogun, Brian T. Bateman.

**Methodology:** Elisabetta Patorno, Brian T. Bateman.

**Project administration:** Elisabetta Patorno, Brian T. Bateman.

**Supervision:** Elisabetta Patorno, Sonia Hernandez-Diaz, Krista F. Huybrechts, Brian T. Bateman.

**Writing – original draft:** Elisabetta Patorno, Sonia Hernandez-Diaz, Krista F. Huybrechts, Brian T. Bateman.

**Writing – review & editing:** Elisabetta Patorno, Sonia Hernandez-Diaz, Krista F. Huybrechts, Rishi J. Desai, Jacqueline M. Cohen, Helen Mogun, Brian T. Bateman.

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
