## [Decision Letter · Decision Letter 0]

30 Jan 2020

Dear Dr. Patorno,

Thank you very much for submitting your manuscript "Gabapentin in Pregnancy and the Risk of Neonatal and Maternal Outcomes" (PMEDICINE-D-19-03278) for consideration at PLOS Medicine. 

[LINK]

In light of these reviews, I am afraid that we will not be able to accept the manuscript for publication in the journal in its current form, but we would like to consider a revised version that addresses the reviewers' and editors' comments. Obviously we cannot make any decision about publication until we have seen the revised manuscript and your response, and we plan to seek re-review by one or more of the reviewers. 

We expect to receive your revised manuscript by Feb 13 2020 11:59PM. Please email us (plosmedicine@plos.org) if you have any questions or concerns.

We look forward to receiving your revised manuscript. 

Sincerely,

Adya Misra

Senior Editor

PLOS Medicine

On behalf of 

Caitlin Moyer, Ph.D.

Associate Editor 

PLOS Medicine

plosmedicine.org

1.Data Availability Statement: Thank you for your willingness to include all relevant data within the manuscript and supporting information files. As noted by reviewer 3, the data are not present in their entirety. Please modify your statement for the data used in your study: 

2. Prospective Analysis Plan: Did your study have a prospective protocol or analysis plan? Please state this (either way) early in the Methods section.

c) In either case, changes in the analysis—including those made in response to peer review comments—should be identified as such in the Methods section of the paper, with rationale

3. Title: Please revise your title according to PLOS Medicine's style. Your title must be nondeclarative and not a question. It should begin with main concept if possible. "Effect of" should be used only if causality can be inferred, i.e., for an RCT. Please place the study design ("A randomized controlled trial," "A retrospective study," "A modelling study," etc.) in the subtitle (ie, after a colon).

4. Abstract: Please structure your abstract using the PLOS Medicine headings (Background, Methods and Findings, Conclusions).

5. Abstract: Please include the study design, population and setting, years during which the study took place,and length of follow up.

6. Abstract: Methods and Findings: Please list some examples of the 70 confounders most critical to the study.

7. Abstract: Methods and Findings: Of the 1,753,865: please also clarify numbers for the unexposed (reference) group.

8. Abstract: Methods and Findings: For all results quantified regarding risk associated with exposure, please present both the 95% CIs and p values.

9. Abstract: Methods and Findings: Please clarify if results presented are for those exposed both early and late in pregnancy, or if these are combined effects of early and late gabapentin exposure on PTB, SGA, and NICUa, and if so, please present separate results for early vs. late exposure (with 95% CIs and p values).

10. Abstract: Methods and Findings: In the last sentence of the Abstract Methods and Findings section, please describe the main limitation(s) of the study's methodology.

11. Abstract: Conclusions: Please address the study implications without overreaching what can be concluded from the data; the phrase "In this study, we observed ..." may be useful. 

12. Author Summary: At this stage, we ask that you include a short, non-technical Author Summary of your research to make findings accessible to a wide audience that includes both scientists and non-scientists. The Author Summary should immediately follow the Abstract in your revised manuscript. This text is subject to editorial change and should be distinct from the scientific abstract. Please see our author guidelines for more information: https://journals.plos.org/plosmedicine/s/revising-your-manuscript#loc-author-summary

13. Methods: Thank you for your note (1st paragraph under “Source of Data and Study Population”) that the details of building the study cohort were previously published. Please however provide some relevant details, in particular which US states were not represented, and the fact that LMP was estimated (and how) as it seems like this could potentially impact interpretations regarding pregnancy timepoints.

14. Methods: How was race/ethnicity defined and by whom? Why was race/ethnicity considered important in this study and what it is believed to represent?

15. Methods: Sensitivity analyses: In the first scenario, you restrict analyses to greater or equal to two gabapentin prescriptions filled during each period to reduce the potential for exposure misclassification. Can you please clarify what is meant by exposure misclassification more explicitly? In the discussion, the interpretation is that such individuals may use gabapentin more consistently.

16. Results: Please note the comment from Reviewer 3 regarding the presentation of the unadjusted results in the text. If presenting both, please consistently describe the adjusted or unadjusted in the text for all outcomes. In the tables, please present both the unadjusted and PS-adjusted results for all measures.

17. Results: As also alluded to by a reviewer, please revise the final sentence of the first paragraph under “Absolute and relative risks of neonatal and maternal outcomes” to: “The relative risks were no longer statistically significant after PS adjustment…”

18. Results: For relative risks associated with all congenital and cardiac malformations, please provide both the 95% CIs and p values associated.

19. Results: In the second paragraph under “Absolute and relative risks of neonatal and maternal outcomes”: please revise to “In PS-adjusted analyses, there was no statistically significant increase in risk of preeclampsia regardless…” Please also present the risk with 95% CIs and p values associated. For associations with preterm birth, small for gestational age, and NICU admissions, please also present the p values for the PS-adjusted findings.

20. Results: Under “Sensitivity, secondary and post-hoc analyses”, please remove the word meaningful and revise the sentence to read: “No significant associations between gabapentin exposure and individual non-cardiac malformations were identified, and elevated point estimates observed for central nervous system defects, ear anomalies, and non-cardiac vascular defects were not statistically significant (eTable6).”

21. Discussion: Please clarify or remove the word “meaningful” from the following sentence (second paragraph, on page 12): “...allowed us to rule out meaningful increases in the risk of overall major malformations among pregnancies exposed to gabapentin during the first trimester with higher precision…”

22. Discussion: Please present and organize the Discussion as follows: a short, clear summary of the article's findings; what the study adds to existing research and where and why the results may differ from previous research; strengths and limitations of the study; implications and next steps for research, clinical practice, and/or public policy; one-paragraph conclusion. Specifically, a discussion of implications and next steps relevant for clinical practice is missing.

23. Conclusions: Please revise the first sentence of the conclusion to temper the implications that can be drawn, we suggest: “Results from this study suggest that gabapentin exposure during early pregnancy does not appear to be associated with teratogenic effects.” or similar.

24. Tables: Tables should be numbered 1-3 rather than 1, 3, and 4.

25. Table 1: Please define the abbreviation for “SD” in the legend.

26. Table 3: Please define abbreviation for T1. Please provide p values in addition to 95% CIs for all outcomes.

27. Table 4: Please define abbreviations for T1, Rx, dx. Please provide p values in addition to 95% CIs for all outcomes. Please also provide the unadjusted results.

28. Figure 2: Please define Pc0.

29. eTable3, eTable4: Please define abbreviation for SD.

30. eTable5: Please define “2 Rx” in the legend. Please provide p values in addition to 95% CIs.

31.eTable6: Please define “2 Rx” and T1 in the legend. Please provide p values in addition to 95% CIs.

32. eTable 7: Please define “T1” in the legend. Please provide p values in addition to 95% CIs. Please also provide the results of the unadjusted analyses.

33. eFigure1: Please define the abbreviation for “RR” in the legend.

34. References: Please use square brackets for in-text citations, like this: [1]. Please use the "Vancouver" style for reference list formatting, and see our website for other reference guidelines: https://journals.plos.org/plosmedicine/s/submission-guidelines#loc-references

35. Checklist: Please ensure that the study is reported according to the STROBE guideline, and include the completed STROBE checklist as Supporting Information. When completing the checklist, please use section and paragraph numbers, rather than page numbers. Please add the following statement, or similar, to the Methods: "This study is reported as per the Strengthening the Reporting of Observational Studies in Epidemiology (STROBE) guideline (S1 Checklist)."

Comments from the reviewers:

Reviewer #1: Thank you very much for allowing me to review this article on the use of Gabapentin in Pregnancy and the Risk of Neonatal and Maternal Outcomes. This is an important topic where safety information is lacking to support clinical management and policy decision making. The article is well written and the approach well reported thereby aiding transparency that is commendable. Please find my comments below for consideration that mainly relate to the methods and results section. 

1) "Continuous Medicaid eligibility was also required among the linked infants for a minimum of three months after birth". 

Could the authors describe why 3 months following birth was chosen to ascertain outcomes in neonates. Because there is an increased risk of pre-term birth with gabapentin-exposed patients meaning the period for outcome detection in infants born at term will be 40 weeks +12 weeks compared to e.g. pre-term 36 weeks +12 weeks, what is the possibility of differential outcome ascertainment of congenital anomalies in offspring? Is there an increased likelihood for example of detecting a cardiac defect if offspring are more likely to be admitted to the NICU etc.?

2) "Pregnancies with exposure to acknowledged teratogenic agents during the first trimester were excluded"

A minor point but there is a risk in misunderstanding this sentence as meaning the exclusion of all teratogenic agents. I suggest clarifying that certain teratogens were still included but handled differently (i.e. some anticonvulsants).

3) The first trimester is defined as the first 90 days of pregnancy. Exposure early in pregnancy is defined as the first 140 days of pregnancy. By definition all exposed individuals in the first trimester should appear in the exposure early in pregnancy cohort (the latter of which is likely to be the larger of the two). The results however, report the number of exposed patients in the first trimester as 4,642 and the number of patients exposed early in pregnancy as 3,745, which seems implausible if I have correctly understood the proposed definition. Please could the authors clarify.

4) A minor point but for clarity in the covariates section of the methods I would add in brackets what indications for gabapentin are included. 

5) It is quite common that some lifestyle information is missing in electronic databases. Was there any missing data on smoking and lifestyle information? If so, how was this handled?

6) "For the primary analyses, we measured maternal comorbidities and concomitant medication use during the 3 months before pregnancy through the end of the first trimester."

Data on maternal comorbidities and concomitant medication were used to estimate the propensity for treatment with gabapentin. However, the propensity score is a subject's probability of treatment, conditional on observed baseline covariates. Including covariate information up to the end of the first trimester means post-baseline data can be used to estimate the propensity score. In many instances gabapentin may have been prescribed before or early in pregnancy it is possible that this population may have more baseline covariate information. What proportion of covariate information included in estimating the propensity score was present at the start of pregnancy between the groups and how might this affect the propensity score distributions? Could the authors discuss potential bias implications in using this approach?

7) "We trimmed the population whose PS fell within the non-overlapping areas of the PS distributions, and created 50 PS-strata according to the distribution of the exposed pregnancies."

Could a figure demonstrating the degree of overlap in the PS distributions be reported to help better understand how generalizable the findings may be?

8) The article reports some statistically significant associations between the use of gabapentin and cardiac malformations. The study also evaluates the risk of maternal outcomes using a cohort consisting of exposed late in pregnancy. Given this cohort of exposed late in pregnancy has already been established could the authors consider using it to test for potential residual confounding in the association with cardiac malformations if gabapentin exposure late in pregnancy may act as a negative control?

9) It may be interesting to examine whether the association with preterm delivery is consistent or varies by the main indications (i.e. epilepsy vs. chronic pain) in case the associated problems with chronic pain and its management influence the timing of delivery. 

Reviewer #2: This paper tackles an extremely important issue, as the use of gabapentin during pregnancy has increased dramatically over recent years.

This is an excellent paper examining if gabapentin exposure during pregnancy increases the risk of neonatal maternal outcomes. The methodology is well thought out, with many additional sensitivy analyses included to confirm their results and is very carefully described in the paper. I really enjoyed reading this paper and can think of no imporvemens to it. It is an exceptional piece of work.

Reviewer #3: This study of the risks associated with gabapentin use during pregnancy is based on a retrospective analysis of a large claims database and on propensity score stratification to control for confounders when comparing women exposed vs not exposed to gabapentin. I found the methods appropriate and well described including a range of sensible sensitivity analyses. I also found the discussion and conclusion well written and reflecting the uncertainty due to potential residual confounding and multiplicity. I only have minor comments, mostly aimed at clarifying the presentation, listed below:

* Please indicate the design of the study in the title. For example: "Gabapentin in Pregnancy and the Risk of Neonatal and Maternal Outcomes: A Propensity-Matched retrospective Cohort Study"

* The data statement suggests that all data necessary to replicate the analysis is available without restriction; however, this does not seem to be the case. Only aggregate results are available in the supplement but not raw data. Given the nature of the data (US Claims data), I suspect that the authors would not have the authority to make the raw data available. This needs to be clarified.

* First trimester (T1) exposure (90 days) and early pregnancy exposure (first 140 days) appear very similar and I wonder whether we need them both. Is it mainly because early pregnancy exposure excludes exposures after day 140 while T1 exposure does not exclude exposure after Day 90. Please confirm and clarify the rationale.

* Please also clarify why exposure is limited to Day 245 and not later. In general, I would like to better understand the rationale for the different cutoffs chosen to define the exposure periods.

* Given how closely the exposure periods overlap, please clarify for selecting different outcomes for T1 exposure vs early pregnancy exposure.

* For the baseline characteristics listed in Table 1, eTable 3 and eTable 4, please clarify the corresponding measuring period when not clear i.e. in particular, please indicate whether the characteristic is measured before the pregnancy and, if so, the length of the "recall" period (e.g. in the 3 months preceding pregnancy) or during pregnancy or both. I note that this is quite well defined in the methods; however, adding details to the tables themselves might help.

* In the results, I would suggest not reporting the unadjusted results as we know that before PS adjustments, the groups are very different and the comparisons potentially misleading. I am not sure I would include unadjusted results in the tables unless the goal is to demonstrate the importance of adjusting. 

* Results section, absolute and relative risks of neonatal and maternal outcomes: In the sentence "The relative risks were no longer meaningfully elevated after PS adjustment [RR, 1.07 (0.94-1.21); RR, 1.12 (0.89-1.40)]." I would suggest changing the wording to "statistically significant" instead of "meaningfully elevated" and clarify what PS "adjustment" mean by using more specific terminology e.g. "PS stratification and weighting".

* Table 4 includes many numbers, which makes it difficult to identify patterns (e.g. significant or inconsistent results). I would suggest using a forest plot instead of a table. This applies to some of the tables in the supplement too.

* Please consider including plots showing the propensity score distribution before adjustment in the supplement.

* When looking at the effect of gabapentin according to dose tertiles, please clarify whether each tertile has been separately adjusted using a new propensity score. To allow each tertile to be compared to the unexposed group, I would expect that a separate propensity score be calculated for each tertile as the propensity is likely to change as the dose increase. Please confirm that this is the approach used. An potential alternative to study a dose-response relationship is to perform an analysis according to tertiles of the propensity score itself.

* Is it possible to add confidence bands to Figure 2 and eFigure 1?

* Please include the STROBE checklist as well as the statistical analysis plan in the supplement.

-Laurent Billot

Reviewer #4: General comments

The authors present a large register-based study on the use of gabapentin in pregnancy and risk of congenital malformations and other important pregnancy and neonatal outcomes.

This author group has strong and well-documented credentials with the field as witnessed by this manuscript. The study is compelling in its scope, execution and reporting. This manuscript substantially advances knowledge on the use of gabapentin in pregnancy and provide much needed decision support to physicians and pregnant women.

That being said, there are some minor issues that I believe should be clarified, justified and discussed prior to publication. 

Specific comments

Introduction

Short sharp and to the point. The sentence "As gabapentin actively crosses…" is trivial (all but very few drugs cross placenta, and I am not sure if the term "actively" is accurate) could be omitted with no loss of information.

Methods

Well described. I would like clarification/justification/discussion of the following points:

a) Exposure window. The authors have chosen to eliminate the 30-day window prior to conception. This is fine, but I suggest they discuss the possible consequences thereof and elaborate on the typical prescription coverage of gabapentin (1 month? Three months?)

b) Exclusion criteria. The authors exclude some known teratogenic drugs. However, some quite notable drugs are omitted, especially ACE-inhibiting drugs and antiepileptics with documented increased risk of major congenital malformations: valproic acid, carbamazepine and phenytoin (I am aware that these latter exposures are included in the PS model). The authors should discuss.

c) Exposure definition. Please state the definition of the start of pregnancy explicitly (LMP?).

d) Outcome assessment. Please discuss the possible consequences of a short follow-up (3 months). Some - especially cardiac - malformations are detected later in the first year after birth.

e) Covariates. While I am confident that the following has been elaborately published elsewhere, I believe that this information is quite important for the reader to have at hand as these covariates are important: 

Alcohol, overweight and smoking: Please briefly describe how these were assessed and stratified and list the % of missing data on these covariates in the original MAX dataset. 

f) Inferential analysis for risk of preterm birth. I believe the authors should discuss the possibility of immortal time bias in this context.

g) Bias analysis (smoking). I acknowledge the reference to the assumed Pc0 prevalence of 30% maternal smoking, but such seem extremely high to me still. Please elaborate. 

Results

Clearly presented if a somewhat long narrative section.

Discussion

Clear and well substantiated in the actual findings. Some adjustments may be warrented pending the response to the issues above.

[LINK]

---

## [Decision Letter · Decision Letter 1]

3 Apr 2020

Dear Dr. Patorno,

Thank you very much for submitting your revised manuscript "Gabapentin in Pregnancy and the Risk of Adverse Neonatal and Maternal Outcomes: A Population-based Cohort Study" (PMEDICINE-D-19-03278R1) for consideration at PLOS Medicine. 

Your paper was re-evaluated by a senior editor and discussed among all the editors here. It was also re-reviewed by three reviewers, including a statistical reviewer. The reviews are appended at the bottom of this email and any accompanying reviewer attachments can be seen via the link below:

[LINK]

In light of the remaining comments from Reviewer #3, I am afraid that we will not be able to accept the manuscript for publication in the journal in its current form, but we would like to consider a revised version that addresses the reviewers' and editors' comments, particularly the points raised by Reviewer #3 regarding the propensity score distributions. Obviously we cannot make any decision about publication until we have seen the revised manuscript and your response, and we plan to seek re-review by one or more of the reviewers. 

We expect to receive your revised manuscript by Apr 10 2020 11:59PM. Please email us (plosmedicine@plos.org) if you have any questions or concerns.

We look forward to receiving your revised manuscript. 

Sincerely,

Caitlin Moyer, Ph.D.

Associate Editor 

PLOS Medicine

plosmedicine.org

1. Abstract (and throughout): Thank you for your responses to this query in the previous round of revision, and we appreciate your willingness to consider the inclusion of p values for your results. Please provide p values in addition to 95% CIs.

2. Author summary: Thank you for including an author summary. Please also include the section: “What Do These Findings Mean?” (In this section, authors should reflect on the new knowledge generated by the research and the implications for practice, research, policy, or public health. Authors should also consider how the interpretation of the study’s findings may be affected by the study limitations.) Please see our author guidelines for more information on formatting this section: https://journals.plos.org/plosmedicine/s/revising-your-manuscript#loc-author-summary

3. Methods: “Source of Data and Study Population”: Please provide a reference for the US Medicaid Analytic eXtract (MAX) database used.

4. Methods: “Source of Data and Study Population”: Thank you for including the STROBE checklist. Please revise the last sentence to read: This study is reported as per the Strengthening the Reporting of Observational Studies in Epidemiology (STROBE) guideline (S1 Checklist)." or similar.

5. Results section, throughout: We thank you for your responses on this subject, but please do report p values to accompany the 95% CIs presented for all results. In instances where you report “significant” associations, please indicate whether the intended meaning is “statistical significance”.

6. Results: “Absolute and relative risks of neonatal and maternal outcomes”: Please provide p values along with 95% CIs for all presented analyses, unadjusted and adjusted, throughout this section. We suggest you change “the relative risks moved” to “the relative risks attenuated” or similar in the sentence “After PS adjustment, the relative risks moved to RR, 1.07 (0.94-1.21) and RR, 1.12 (0.89-1.40), respectively.” to enhance clarity.

7. Results “Sensitivity, secondary and post-hoc analyses”: Please provide p values along with 95% CIs for all presented analyses.

8. Results: (and response to editor comment #16): Thank you for your response to this, please do report (in the table) the unadjusted results for the secondary analyses.

9. Figure 2: In the title, please include panels E and F (...unexposed and gabapentin exposed women in either late (A, C, E) or both early and late in pregnancy (B, D, F)), or remove the panel IDs from the title as this information is found in the legend.

10. eFigure 1: Please use a consistent number of decimal places for the x axes across panels. In the legend, please provide a description of the comparison groups: Exp_Group: exposure group; Gaba_Group1-4: gabapentin exposed women in exposure groups 1 to 4 (presumably gaba = gabapentin, exposure groups 1-4 correspond to time periods of exposures). Please include a descriptive figure legend, if possible.

11. eTable 5: Please provide p values.

References: Please check the formatting for reference 3, removing the trademark symbol. Please double check that your references use the "Vancouver" style for reference formatting (it looks like you may need a period between the journal names and the publication year), and see our website for other reference guidelines https://journals.plos.org/plosmedicine/s/submission-guidelines#loc-references

12. Checklist: Thank you for providing your STROBE checklist. Please revise the checklist, using sections and paragraphs, rather than page numbers, to refer to locations throughout the manuscript.

Comments from the reviewers:

Reviewer #1: Thank you for asking me to review the revision. The responses are satisfactory. I am clearer that a patient's complete claim history prior to the pregnancy is not available. For this reason information accruing throughout the first trimester is the only way to identify the necessary information. Although it would normally be expected that key past medical history is recorded early on, I agree that more medical conditions could have chance to accumulate over time. The primary propensity score analysis was estimated only on the basis of chronic conditions or treatments that are not expected to be causal intermediates between gabapentin and pregnancy complications, which is valid and the variables are listed in etable 3. Assuming the high dimensional propensity score (HDPS) uses more covariates than simply those in etable 3, the HDPS could be more at risk from the appearance of conditions and treatments after gabapentin exposure has commenced. However, this is a secondary analysis and given the effect estimates are similar to those of the primary PS analysis it seems unlikely to be an issue here. 

Reviewer #3: Looking at eFigure1 which shows the overlap in PS distribution, it looks as if 100% (or very close to 100%) of the unexposed group fell within the first bar of the histogram whereas, for the exposed groups, the proportion included in the first bar varied between approximately 65% and 92%. This indicates a substantial lack of overlap in the propensity score distributions between the exposed and non-exposed groups, yet the authors report retaining 100% (or 99.8%) of exposed cases.

Please clarify the rule used for trimming and the reason for having so few cases trimmed. This lack of PS overlap suggests that a number of exposed cases would have been matched (i.e. included in the same strata) as non-exposed cases who had a much lower propensity score. I wonder about the possible biases induced by including exposed cases with high (non-overlapping) propensity scores and whether a sensitivity analysis excluding exposed cases who have a "non-overlapping" propensity score might alleviate those concerns.

I note that the authors do not wish to report p-values. While p-values can indeed be misinterpreted, I feel that they help quantify the degree of consistency between the observed data and the hypotheses and are a useful complement to confidence intervals. In fact, the articles cited, e.g. the ASA statement, do not necessarily ask authors to avoid using p-values but rather warn against misinterpretation. For the sake of transparency and interpretation, I would encourage the reporting of p-values in addition to confidence intervals but will leave the decision with the editors.

Reviewer #4: The authors have satisfactorily addressed issues raised in my review and revised the manuscript accordingly.

[LINK]

---

## [Decision Letter · Decision Letter 2]

6 May 2020

Dear Dr. Patorno,

Thank you very much for submitting your revised manuscript "Gabapentin in Pregnancy and the Risk of Adverse Neonatal and Maternal Outcomes: A Population-based Cohort Study" (PMEDICINE-D-19-03278R2) for consideration at PLOS Medicine. 

Your paper was re-evaluated by a senior editor and discussed among all the editors here. It was also discussed with an academic editor with relevant expertise, and sent to a statistical reviewer for re-review. The reviews are appended at the bottom of this email and any accompanying reviewer attachments can be seen via the link below:

[LINK]

In light of the outstanding editorial issues, I am afraid that we will not be able to accept the manuscript for publication in the journal in its current form, but we would like to consider a revised version that addresses the editors' comments. In particular, please note that we cannot move forward with your manuscript without the inclusion of p-values. Obviously we cannot make any decision about publication until we have seen the revised manuscript and your response.

In revising the manuscript for further consideration, your revisions should address the specific points made by the editors. Please also check the guidelines for revised papers at http://journals.plos.org/plosmedicine/s/revising-your-manuscript for any that apply to your paper. In your rebuttal letter you should indicate your response to the editors' comments, the changes you have made in the manuscript, and include either an excerpt of the revised text or the location (eg: page and line number) where each change can be found. Please submit a clean version of the paper as the main article file; a version with changes marked should be uploaded as a marked up manuscript.

We expect to receive your revised manuscript by May 13 2020 11:59PM. Please email us (plosmedicine@plos.org) if you have any questions or concerns.

We look forward to receiving your revised manuscript. 

Sincerely,

Caitlin Moyer, Ph.D.

Associate Editor 

PLOS Medicine

plosmedicine.org

1. Response to reviewers/editors: Thank you for the responses to the reviewer and editor comments. 

- Response to Editorial Request #1:“We have asked on previous occasions for p values to be provided alongside 95% CIs. These are still not provided and I should advise you that we will be unwilling to continue with this paper unless you provide them. We understand the limitations of presenting and interpreting p-values that you have outlined in your response letter. However, our readers expect to see them, and thus PLOS Medicine requires both p-values and 95% CIs when presenting results. We will withdraw the paper from our files if these are not provided on the resubmission

- Response to Editorial Request #8: Please present the results from the unadjusted analyses in addition to the adjusted analyses as requested previously (in a separate table would be acceptable).

2. Title: Please include the population and setting (country)- we suggest “Gabapentin in pregnancy and the risk of adverse neonatal and maternal outcomes: A population-based cohort study of the US Medicaid Analytic eXtract (MAX) dataset”

3. Abstract: Methods and Findings: Please include some summary demographic information on the individuals included in the dataset and please include the months along with the years for the included pregnancies.

4. Abstract: Methods and Findings: Please provide p-values in addition to 95% CIs.

5. Abstract: Methods and Findings: Can you please clarify this sentence: “There was no association with preeclampsia after adjustment” to indicate which relationship you are describing?

6. Results: Last sentence of “Study cohort and patient characteristics”- does the superscript 19 indicate reference 19? If so please place the reference in square brackets: [19].

7. Results: “Absolute and relative risks of neonatal and maternal outcomes”: Please provide p-values associated with the risk ratios and 95% CIs here, and throughout the manuscript. In the methods, please specify the significance level used (eg, P<0.05, two-sided) and the statistical test used to derive a p-value.

8. Results: “Sensitivity, secondary and post-hoc analyses”: For the following sentence, please clarify whether the risk was significantly increased for the subgroup of patients with an epilepsy/seizure diagnosis: “Of note, the risk of cardiac malformations in pregnancies exposed to gabapentin during T1 was significantly increased when we re-defined the exposure based on ≥2 filled prescriptions [RR, 1.40 (1.03-1.90)], and remained elevated in a post-hoc analysis with hdPS adjustment [RR, 1.40 (1.03-1.90)], and in a subgroup analysis that restricted to patients with a recorded diagnosis of epilepsy or seizures [RR, 1.40; 0.73-2.71)].”

9. Discussion: Last sentences of paragraph beginning “This study has limitations”: We suggest revising to: “Our results add to the current understanding of the safety of gabapentin prenatal use and provide pregnant women with pain conditions and epilepsy and their providers with important information, which can guide clinical decisions during pregnancy. Our findings also suggest that pregnant women using gabapentin during pregnancy may be considered for targeted interventions to monitor for and promptly respond to the potential adverse outcomes associated with the use of this agent.”

We also suggest that the discussion of the strengths and limitations of the study be a separate paragraph from the discussion of implications and next steps for research.

10. Conclusions: The caveat regarding the impact of multiple analyses on the cardiac finding would be more appropriately discussed in the paragraph describing the study’s limitations. “A moderate increase in the risk of cardiac malformations – in particular conotruncal defects – cannot be excluded, although in the context of multiple analyses the possibility of a chance finding should be taken into consideration.” In the last sentence, please change “clinicians need to” to “clinicians should” or similar.

11. Table 2 and Table 3, and eTable 5, 6, and 7: Please provide p values associated with these results.

Comments from the reviewers:

Reviewer #3: My previous query about the apparent lack of overlap in PS distribution has been adequately addressed. I have no further comment.

-Laurent Billot

[LINK]

---

## [Editor Report · Decision Letter 3]

17 Jul 2020

Dear Dr. Patorno,

Thank you very much for re-submitting your manuscript "Gabapentin in Pregnancy and the Risk of Adverse Neonatal and Maternal Outcomes: A Population-based Cohort Study" (PMEDICINE-D-19-03278R3) for review by PLOS Medicine.

After discussing the paper with my colleagues, I am pleased to say that provided the remaining editorial and production issues are dealt with we are planning to accept the paper for publication in the journal.

[LINK]

We look forward to receiving the revised manuscript by Jul 24 2020 11:59PM. 

Sincerely,

Caitlin Moyer, Ph.D.

Associate Editor 

PLOS Medicine

plosmedicine.org

Requests from Editors:

1.Title: Thank you for updating your title to: “Gabapentin in Pregnancy and the Risk of Adverse Neonatal and Maternal Outcomes: A Population-based Cohort Study nested in the US Medicaid Analytic eXtract Dataset.” Please also make sure this title is updated on the manuscript submission form/system to ensure that the title is carried forward accurately.

2.Author summary: "Why was this study done?": Please remove the word “only” from the second bullet point.

3.Discussion: Conclusion paragraph: Please revise this sentence as follows, changing “increased” to “higher” to be consistent with wording elsewhere in the text: “Maternal use of gabapentin, particularly late in pregnancy, was associated with a higher risk of preterm birth, SGA, and NICU admission; an association that was only partially explained by confounders.”

4.Sections titled “Author contributions”, “Funding”, “Competing interests”, and “Data availability statement”: These can be removed from the manuscript body; please be sure the information is entered into the relevant place in the manuscript submission form and it will automatically be included with the manuscript.

5.Appendix: Please include each document/table as a separate supporting information file.

[LINK]

---

## [Editor Report · Decision Letter 4]

30 Jul 2020

Dear Dr Patorno, 

On behalf of my colleagues and the academic editor, Dr. Sarah J Stock, I am delighted to inform you that your manuscript entitled "Gabapentin in Pregnancy and the Risk of Adverse Neonatal and Maternal Outcomes: A Population-based Cohort Study nested in the US Medicaid Analytic eXtract Dataset" (PMEDICINE-D-19-03278R4) has been accepted for publication in PLOS Medicine. 

PRODUCTION PROCESS

PRESS

PROFILE INFORMATION

Thank you again for submitting the manuscript to PLOS Medicine. We look forward to publishing it. 

Best wishes, 

Caitlin Moyer, Ph.D.

Associate Editor 

PLOS Medicine

plosmedicine.org